chemical engineering/energy/materials science

silicalite-1, ZSM-5 zeolites, crystal size, methanol to aromatics

**Author for correspondence:**
Xianjun Niu
e-mail: xjniu1984@163.com

This article has been edited by the Royal Society of Chemistry, including the commissioning, peer review process and editorial aspects up to the point of acceptance.

# Size controllable synthesis of ZSM-5 zeolite and its catalytic performance in the reaction of methanol conversion to aromatics

Xianjun Niu, Yang Bai, Yi-en Du, Hongxue Qi and Yongqiang Chen

Department of Chemistry and Chemical Engineering, Jinzhong University, Jinzhong, Shanxi 030619, People's Republic of China

XN, 0000-0002-5019-9586; HQ, 0000-0002-0728-4312; YC, 0000-0002-7879-896X

ZSM-5 zeolites were hydrothermally synthesized with commercial silica sol, and the crystal size was controlled by adding silicalite-1 seed in the synthetic system. The crystal size of ZSM-5 was affected by the crystallization time of seed, seed content and seed size. ZSM-5 zeolites with controllable particle size in the range of 200–2200 nm could be obtained. The prepared samples with different particle sizes were used for the reaction of methanol conversion to aromatics (MTA). The results suggested that the HZSM-5 catalyst with small crystal size showed much longer catalyst lifetime and higher selectivity for $C_5^+$ hydrocarbons and aromatics, especially $C_9^+$ aromatics in the MTA reaction. Moreover, the introduction of zinc (Zn) into the HZSM-5 zeolites can considerably promote the selectivity to aromatics in the products. Zn modified HZSM-5 zeolites with different Zn loading amounts were prepared by the incipient wetness impregnation method, and the highest aromatics selectivity was obtained when the Zn loading was 1.0%. The improvement of methanol aromatization was ascribed to the synergistic effect of Brønsted acid sites and the newly formed Zn-Lewis acid sites.

## 1. Introduction

Aromatic hydrocarbons are one of the most important basic raw materials for the production of petrochemical products. The industrial production processes of aromatics including naphtha hydrogenation, reforming or cracking gasoline, are almost always based on petroleum processing. However, the limited crude oil

resource and the probable decline in oil supply make the price volatile. Therefore, it is crucial to develop new processes as an alternative to the synthetic route to produce aromatics or other chemicals [1–8].

The reaction of methanol to hydrocarbons (MTHs) over a zeolite catalyst discovered in the 1970s is a non-petroleum route for the production of various hydrocarbons [9–13]. ZSM-5 zeolite is one of the most concerned materials in the MTH reaction. The particle size of the zeolite has a marked impact on catalytic behaviours [14–16]. Compared with the large crystallites, the small-sized ZSM-5 crystallites show a long catalyst lifetime and high selectivity of $C_5^+$ hydrocarbons and aromatics in the MTH reaction [17–19]. On the other hand, although the nano-sized zeolite crystals display excellent performance in anti-coking, a decrease of the hydrothermal stability may also become significant owing to the increase of defect sites [20]. Therefore, to optimize the catalytic performance of zeolites, it is of importance to finely control the crystal size and to understand the effect of crystal size on the catalytic behaviours [21–23]. Nano-sized zeolite crystals have attracted considerable attention in the field of catalysis, and several methods including confined space synthesis, [24,25] low-temperature synthesis, [26,27] and seed-induction synthesis, [18,23,28–31] have been devised to synthesize small-sized zeolite crystals. Nowadays, the seed-induction synthesis method is paid more attention to because of its convenient and effective control of the crystal size of zeolites. Mi *et al.* [32] synthesized nano-ZSM-5 zeolites in the particle size range of 200–1200 nm by controlling the content of template tetrabutylammonium bromide (TPABr) and silicalite-1 seed. The template and seed play a synergistic role in the ZSM-5 crystallization. Lee and co-workers [33] prepared a series of ZSM-5 nanocrystals using this method and found the size of ZSM-5 nanocrystals was influenced by the amount of seeds, crystallization time and Si/Al ratio. Tang and co-workers [29,34,35] used silicalite-1 as seed to prepare the ZSM-5 zeolites with the controllable size of 270–1100 nm without adding the templates. They reported that the growth of ZSM-5 crystal was dependent on the amount of the silicalite-1 seed or seed size, and a crystallization mechanism of seed surface was proposed for the seed-induction synthesis process.

To further improve the catalytic performance of ZSM-5 catalysts, the metal species have been used to modify ZSM-5 zeolites to enhance the selectivity of aromatic hydrocarbons in the reaction of methanol to aromatics (MTA) [36–44]. Among these modified metallic species, zinc (Zn) is most attractive because of the low price and good aromatization performance. Tan and co-workers [45] synthesized mesoporous Zn/ZSM-5 zeolite by directly introducing Zn into the alkali treatment process. With increasing Zn content, the benzene, toluene and xylene (BTX) selectivity of mesoporous Zn/ZSM-5 zeolite was gradually improved owing to the emergence of $ZnOH^+$ species. Ni *et al.* [46] prepared nano-H [Zn, Al] ZSM-5 which was composed of $250 \times 50 \times 25$ nm crystalline aggregates by the hydrothermal synthesis in one step, and the catalyst showed high selectivity of BTX and good catalytic stability. Wei and co-workers [47,48] synthesized nano-ZSM-5 zeolite with the short B-axis, and the corresponding short straight channels effectively inhibited carbon deposition during the MTA process. Although the selectivity of aromatic hydrocarbons was only about 30% over ZSM-5, the selectivity of aromatic hydrocarbons reached more than 90% after the impregnation with Zn. Introducing Zn into ZSM-5 could form the new Lewis acid sites and promote the methanol aromatization ability [13,18,45].

Herein, the ZSM-5 zeolite was hydrothermally synthesized with commercial silica sol, and the particle size was controlled by adding silicalite-1 seeds. ZSM-5 zeolites with controllable particle size in the range of 200–2200 nm could be obtained by adjusting the seed crystallization time, seed amount and seed size and then used as the catalysts in the MTA reaction. The relationship between the particle size and the catalytic activity was investigated. Zn was introduced into ZSM-5 zeolites by the incipient wetness impregnation method to promote the selectivity of aromatics in the products, and the optimum loading content of Zn was confirmed.

# 2. Experimental

## 2.1. Catalyst synthesis

Different-sized silicalite-1 crystals were synthesized by appropriately modifying the procedures reported in [49]. The silicalite-1 seeds synthesized at 60°C and 80°C were denoted as S60 and S80, respectively. Without specifying the crystallization time of the seeds, S60 and S80 respectively, represented the seeds synthesized at 60°C for 20 d and 80°C for 6 d. The prepared colloidal silicalite-1 crystalline mixture that had not undergone any process treatment was directly used as the seeds to synthesize ZSM-5 zeolite. The ZSM-5 zeolite was synthesized with the molar composition: $SiO_2$:0.011 $Al_2O_3$:0.02 $Na_2O$:0.15 TPAOH:30 $H_2O$ prepared from sodium hydroxide (NaOH), silica sol (30 wt.% $SiO_2$),

sodium aluminate (NaAlO$_2$, Al$_2$O$_3$ ≥ 41.0 wt.%) and tetrapropylammonium hydroxide (TPAOH, 25 wt.% in aqueous solution). A calculated number of silicalite-1 seeds was added to the above-mentioned synthetic system and then crystallized at 170°C for 48 h. The solid mixture was centrifuged and washed until the mother solution exhibited a pH value of 7–8. The obtained samples were dried at 110°C for 12 h and calcined at 550°C for 10 h. The HZSM-5 catalysts were prepared through ion-exchanging of Na$^+$ with NH$_4$NO$_3$ solution (1 M, m(liquid)/m(solid) = 4) at 80°C for 4 h and then treated by drying and calcination. The obtained samples were designated as Sx-y-z, in which S stands for the silicalite-1 seed used in the synthesis of ZSM-5, x represents the synthesis temperature (in °C) of the silicalite-1 seed and y and z represents the weight per cent of the seed and the particle diameter of the synthesized ZSM-5 sample in micrometres, respectively.

The Zn species were introduced into HZSM-5 zeolite by the incipient wetness impregnation method, and the different Zn contents modified HZSM-5 zeolites were prepared through varying the concentration of Zn(NO$_3$)$_2$ solution and then dried at 110°C and calcined at 550 for 6 h in air. The samples were denoted as Zn/Sx-y-z.

## 2.2. Catalyst characterization

The X-ray diffraction (XRD) patterns were recorded on a Rigaku MiniFlex II desktop X-ray diffractometer with CuK$\alpha$ radiation to identify the crystalline phases. Crystallinity represents the degree of crystallization perfection of zeolites, and a reference sample is usually selected to compare the degree of crystallization with relative crystallinity. In this work, the relative crystallinity was determined by comparing the total peak areas in the range of $2\theta = 22–25°$ with those of the ZSM-5 zeolite prepared without adding seeds. The amount of Zn, Al and Si in the samples was taken on an inductively coupled plasma-atomic emission spectrometer (ICP-AES, Autoscan16, TJA). Adsorption/desorption of nitrogen was measured on a BELSORP-max instrument. Temperature-programmed desorption of NH$_3$ (NH$_3$-TPD) was performed on a Micromeritics AutoChem II 2920 instrument. The morphology and size of the colloidal silicalite-1 seeds and the HZSM-5 samples were observed using a transmission electron microscope (TEM, JEM-1011) or field emission scanning electron microscope (FESEM, JSM-7001F). Fourier transform infrared (FT-IR) and infrared spectra for pyridine adsorption (Py-IR) were measured on Tensor 27 FT-IR spectrometer.

## 2.3. Catalytic tests

The MTA reaction was carried out in a fixed-bed microreactor (a stainless steel tube with an inner diameter of 10 mm). The products were analysed by gas chromatography. The activation treatment of the catalysts, reaction conditions and products analysis were performed by following the previously described procedures [44]. The methanol conversion and product selectivity were calculated as follows:

$$\text{methanol conversion (\%)} = \frac{n^i_{\text{CH}_3\text{OH}} - n^o_{\text{CH}_3\text{OH}}}{n^i_{\text{CH}_3\text{OH}}} \times 100. \tag{2.1}$$

The $n^i_{\text{CH}_3\text{OH}}$ and $n^o_{\text{CH}_3\text{OH}}$ represented the total amount of methanol feed and the amount of unreacted methanol, respectively:

$$\text{product selectivity (\%)} = \frac{m_i}{m} \times 100. \tag{2.2}$$

The $m_i$ and $m$ represented the mass of product $i$ in the hydrocarbons products and the total mass of all hydrocarbons products, respectively.

# 3. Results and discussion

## 3.1. Characterization of the silicalite-1 seed

After centrifuging, washing and drying, the properties of the solid silicalite-1 crystals were characterized. The electronic supplementary material, figure S1 showed the XRD patterns of silicalite-1seeds crystallized at 60°C for 20 d and 80°C for 6 d. It could be seen that both the seeds possessed Mobil five (MFI) structure [50]. The crystallinity of the seed increased with the increase of crystallization temperature. FT-IR analysis showed that the characteristic bands of the framework in MFI-type zeolites, such as the bending vibration

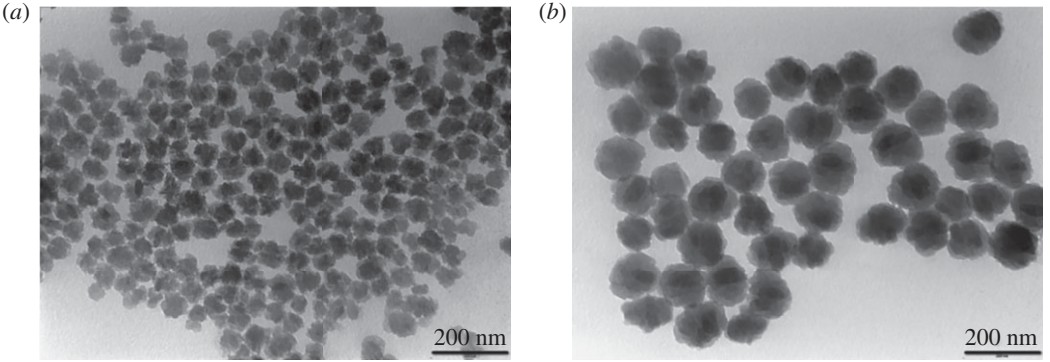

**Figure 1.** TEM images of silicalite-1 seeds. (*a*) S60—seeds synthesized at 60°C for 20 d and (*b*) S80—seeds synthesized at 80°C for 6 d.

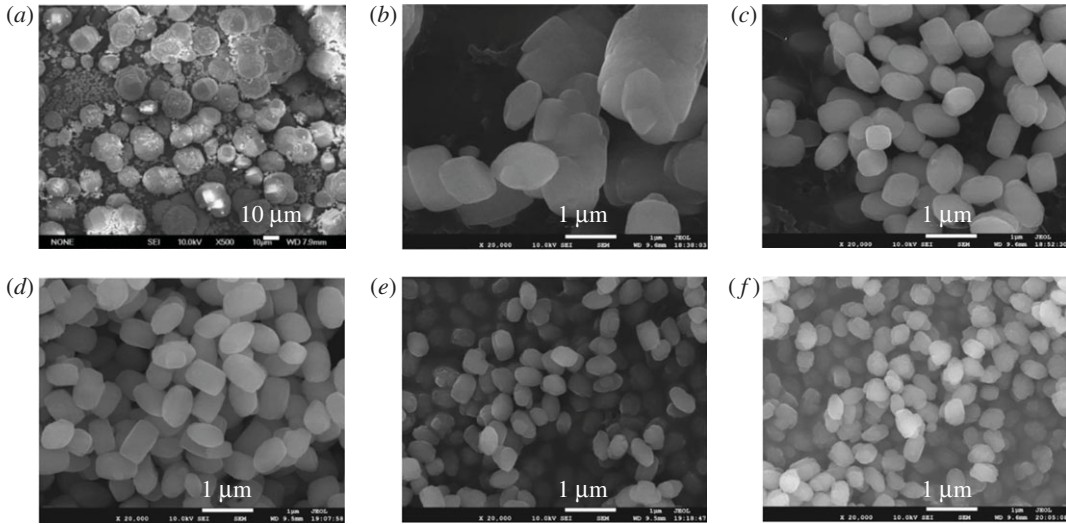

**Figure 2.** SEM images of ZSM-5 synthesized by using 0.2 wt.% S60 crystallized with different times (crystallization time: (*a*) 0 d (no seeds), (*b*) 1 d, (*c*) 3 d, (*d*) 5 d, (*e*) 10 d and (*f*) 20 d).

of Si-O tetrahedra at 450 cm$^{-1}$, the double-ring vibration at 550 cm$^{-1}$ and the stretching vibrations at 1092 and 1225 cm$^{-1}$, all appeared (electronic supplementary material, figure S2) [51]. The TEM images given in figure 1 indicated that the particle size distributions of the silicalite-1seeds were uniform and the particle sizes of S60 and S80 were *ca* 70 and 120 nm, respectively. The particle size of silicalite-1seeds depended on the synthesis temperature when other synthetic conditions were the same.

## 3.2. Effect of the silicalite-1 seed on physico-chemical properties of ZSM-5 zeolites

The ZSM-5 zeolite with uniform and tunable crystal size especially in the scale of nanometer to submicron can be achieved by adding the seed crystals to the synthesis system [29]. The SEM images (figure 2) showed the influences of the seed crystallization time on the crystal size of the synthesized ZSM-5 zeolite. It can be seen that the crystal size of ZSM-5 prepared without adding the seeds exhibited a broad crystal size distribution and a larger particle size (1–30 µm). With the addition of 0.2 wt.% silicalite-1 seeds crystallized for 1 d at 60°C, the crystal size of the prepared ZSM-5 zeolite obviously decreased and was relatively uniform, although the phenomenon of agglomeration was observed. The crystal size of the synthesized ZSM-5 zeolite decreased gradually and became uniform by increasing the crystallization time of the added seeds. The crystal size of ZSM-5 zeolite did not show evident changes when the crystallization time of the used seed exceeded 10 d (figure 2*e,f*). It was reported that the size and population density of the seed primarily determined the size of the synthesized nanocrystals [52]. With prolonging the crystallization time of the seed, the number of crystal nucleus generated gradually increased; the crystal size of the seed grew further and finally

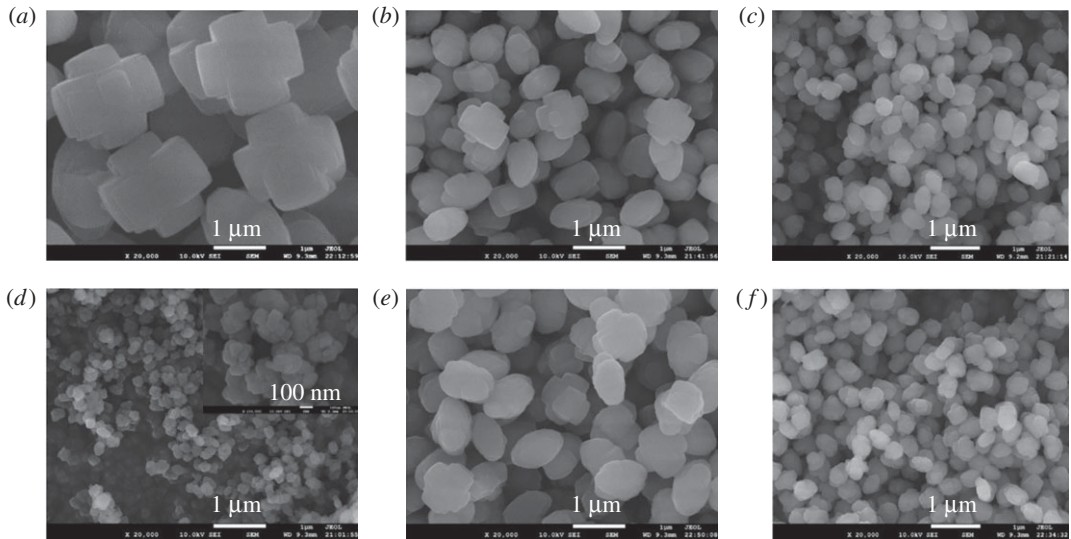

**Figure 3.** SEM images of ZSM-5 synthesized with different amounts and different crystal sizes of silicalite-1 seeds. (a) S60-0.004-2, (b) S60-0.04-1, (c) S60-0.2-0.5, (d) S60-2-0.25, (e) S80-0.2-1 and (f) S80-2-0.5.

became relatively uniform particles with *ca* 70 nm. Therefore, when adding the same amount of the colloidal seeds without any centrifugation or washing process into the synthesis system of ZSM-5 zeolite, silicalite-1 seeds prepared within 10 d had relatively few crystal nucleus and nonuniform particle size, leading to relatively larger particle size and nonuniform in size and distribution of the synthesized ZSM-5 zeolite. More and more silicalite-1 crystal nuclei in the gel solution were formed as the crystallization time of the seed was prolonged. The increase of seed crystal nuclei resulted in the decrease of particle size of the synthesized ZSM-5 zeolite. The size of silicalite-1 seeds tended to be stable, and the amount of nucleus and crystal size of silicalite-1 almost did not change over 10 d, as a result the crystal size of obtained ZSM-5 was uniform in size and distribution [26,53].

Figure 3 gives the SEM images of several synthesized typical samples by adding different contents of S60 and S80. The morphology and particle size of all synthesized samples were uniform. When the same seed was added to the synthesis system, the crystal size of the ZSM-5 zeolites gradually decreased with increasing the amount of the seed (figure 3a–f). The crystal size of the synthesized samples decreased from 2 µm to 0.25 µm by increasing the amount of S60 from 0.004 wt.% to 2 wt.%. The smallest size S60-2-0.25 was aggregated from nanocrystals of tens of nanometers. It can be seen from figure 3c,e that the particle sizes of the synthesized samples by adding S60 and S80 with the same amount of 0.2 wt.% were 0.5 and 1 µm, respectively. In the case of adding the same number of seeds, the crystal size of ZSM-5 synthesized with the small crystal seeds (S60) was smaller than that synthesized with the large crystal seeds (S80) (figure 3c,e and d,f). The phenomenon indicated that both seed size and additive amount significantly affected the crystal size of the synthesized ZSM-5 zeolites. The crystal size of ZSM-5 can be tuned from 200 to 2200 nm through adjusting the amount of the seeds (0.004–10 wt.%) as well as the synthesis condition of the seeds (crystallization temperature and time) (electronic supplementary material, figure S3). Especially in the scale of 200–1000 nm, the crystal size of ZSM-5 can be adjusted within 100 nm by varying the amount and size of the seeds. The maximum uniform particle size of synthesized ZSM-5 was about 2.2 µm with the addition of the seed crystals. It was reported that the particle size of ZSM-5 zeolite can be well controlled in the nanometer to submicron size range by the seed-induction method. Majano *et al*. [54] synthesized nano-sized ZSM-5 from organic template-free gel systems containing silicalite-1 seeds. The size of the ZSM-5 crystals prepared at different temperatures varied from 70 to 700 nm. Tang and co-workers [29] successfully synthesized ZSM-5 zeolites with adjustable submicron-crystal size (270–1100 nm) by the seed-induction method. The addition of crystal seed was beneficial to the formation of ZSM-5 zeolite with smaller particle size because it can accelerate nucleation and inhibit crystal growth [28].

The change of the particle size of ZSM-5 samples with the crystallization time was investigated. In the initial stage, the gel used to synthesize ZSM-5 was almost full of irregular granules with the crystal size of several tenths nanometers which were smaller than the added S80 seeds (figure 4a). This was owing to slight or partial dissolution of the silicalite-1 seeds when added to the alkaline synthetic gel [55]. In an organic template-free gel system, it was also found that the addition of the seeds into a synthetic

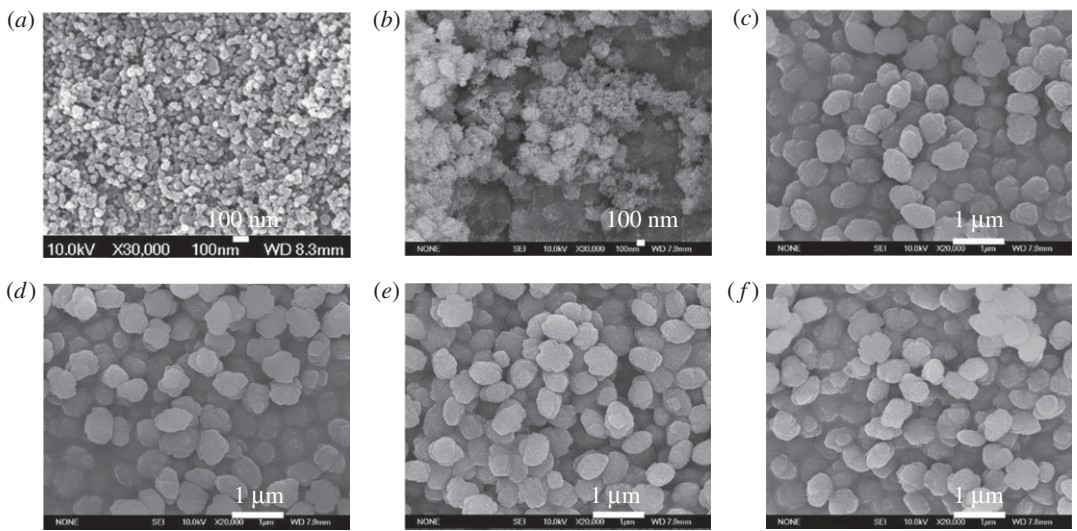

**Figure 4.** SEM images of ZSM-5 synthesized with different crystallization times by using 0.8 wt.% S80. (*a*) 1 h, (*b*) 2 h, (*c*) 2.5 h, (*d*) 5 h, (*e*) 12 h and (*f*) 48 h.

mixture caused deposition of small amorphous nanoparticles formed by depletion of the silicalite-1 gel [29]. With increasing the crystallization time, the amorphous silica and alumina species located around the seeds started to deposit onto the seed surface and agglomerate together to generate ZSM-5 crystals with fluffy round morphology (figure 4*b*). When the crystallization time reached 2.5 h, the final crystals of ZSM-5 were obtained (figure 4*c*). The size and morphology of ZSM-5 remained unchanged when the crystallization time was prolonged (figure 4*d–f*).

Figure 5*a* gives the XRD patterns of the ZSM-5 zeolites synthesized with different crystallization times by using 0.8 wt.% S80. At the initial stage of crystallization (1 h), only partially dissolved seeds with damaged structures existed. Therefore, the peak strength of XRD was very weak. With prolonging the crystallization time, ZSM-5 crystals began to form, and the peak strength obviously increased. Finally, the perfect ZSM-5 crystals were prepared after crystallization for 2.5 h, and the peak intensity hardly changed at all, consistent with above SEM results. The relative crystallinity was determined by comparing the total peak areas in the range of $2 = 22–25°$ with those of the ZSM-5 zeolite prepared without adding the seeds. The decrease of the particle size had little effect on the crystallinity of ZSM-5 zeolites (table 1). In addition, as can be seen from the crystallization curves of ZSM-5 zeolites (figure 5*b*), the addition of the seeds can greatly shorten the crystallization time of ZSM-5. The ZSM-5 zeolite product can be accomplished within 2.5 h in existence of the seeds, whereas it took more than 24 h without adding the seeds.

Figure 6*a* shows the XRD patterns of the prepared ZSM-5 zeolites with different crystal sizes. All the samples prepared with adding silicalite-1 seed exhibited typical peaks of zeolite with MFI structure and possessed high crystallinity (table 1). To quantify the porosity of the obtained ZSM-5 zeolites, $N_2$ adsorption–desorption measurements were carried out, and the adsorption–desorption isotherms are given in figure 6*b*. All the shapes of isotherms were typical of the micropore materials. A steeper increase in the high relative pressure region ($P/P^0 = 0.8–1.0$) of the adsorption and desorption branches was observed for the smallest crystal sample (S60-2-0.25). This may be owing to the mesopores aggregated by intercrystalline voids between the primary nanocrystals, which was consistent with the SEM results [28]. As shown in table 1, the small crystallites possessed higher specific surface area, external surface area and larger pore volume which decreased with the increase of particle size.

Tang and co-workers [29] reported that the growth of ZSM-5 crystal for the seed-induction synthesis process followed a seed surface crystallization mechanism. The ZSM-5 zeolite grew along the surface of the silicalite-1 seed, and its morphology and crystal size were related to the structure of the silicalite-1 seed. Xue *et al.* [28] suggested that the silicalite-1 seeds initially dissolved into smaller basic units in an alkaline synthetic system and then further participated in the induction of ZSM-5 growth. According to the above characteristic results, when the same amount of the seeds was added, the crystal size of ZSM-5 zeolites synthesized by the different sizes of the seeds was different. We prefer the former view that the ZSM-5 zeolite grows on the seed surface. As shown in figure 7, in the beginning, Si and Al species are scattered around the silicalite-1 seeds, and then these Si and Al species aggregate and grow along the surface of the seeds. At the same time, owing to the action of

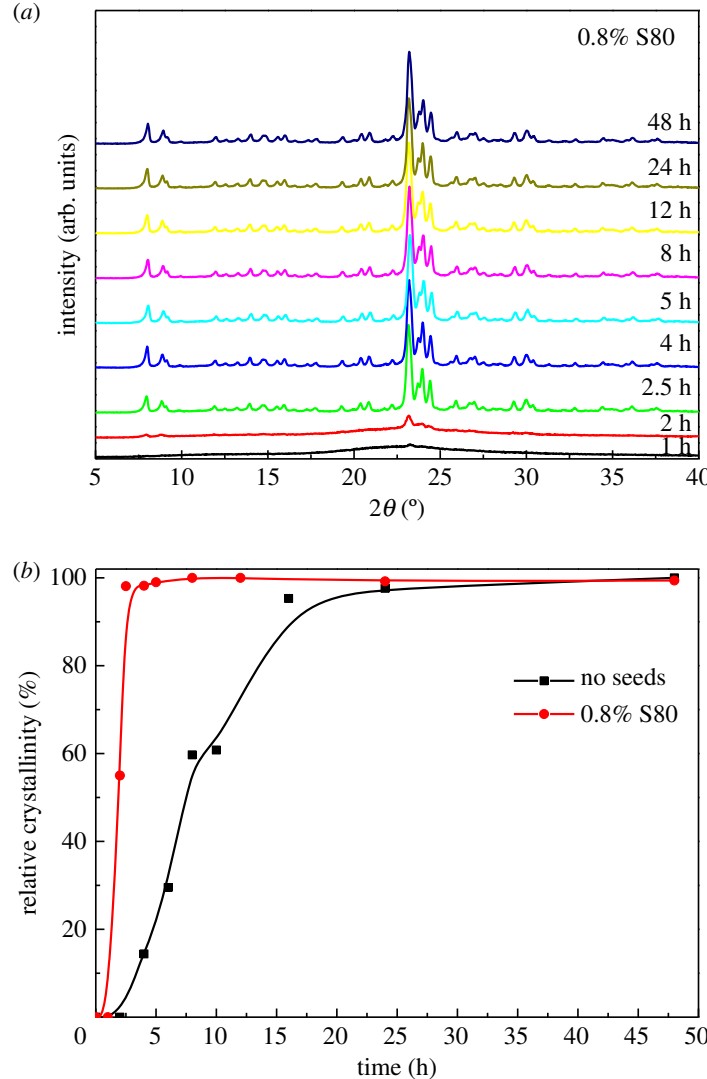

**Figure 5.** (*a*) XRD patterns of ZSM-5 synthesized with different crystallization times by using 0.8 wt.% S80 and (*b*) the crystallization curves of the synthesized ZSM-5 zeolites with or without adding seeds.

**Table 1.** Physicochemical properties of ZSM-5 samples with different crystal sizes.

| samples | Si/Al ratio[a] | $S_{BET}$ (m² g⁻¹)[b] | $S_e$ (m² g⁻¹)[c] | $V_t$ (cm³g⁻¹)[d] | crystallinity (%)[e] | crystal size (µm) |
|---|---|---|---|---|---|---|
| ZSM-5 (no seeds) | 36.01 | 371.14 | 10.96 | 0.21 | 100 | 1–30 |
| S80-0.2-1 | 36.52 | 386.69 | 14.47 | 0.22 | 98.9 | 1.0 |
| S80-2-0.5 | 37.66 | 390.77 | 26.79 | 0.25 | 98.2 | 0.5 |
| S60-0.004-2 | 36.02 | 380.82 | 13.80 | 0.21 | 97.8 | 2.0 |
| S60-0.04-1 | 36.03 | 384.32 | 16.54 | 0.23 | 98.6 | 1.0 |
| S60-0.2-0.5 | 37.33 | 390.11 | 25.05 | 0.26 | 96.8 | 0.5 |
| S60-2-0.25 | 38.64 | 395.01 | 48.81 | 0.53 | 99.6 | 0.25 |

[a]From ICP analysis.
[b]Surface area from Brunauer–Emmett–Teller (BET)-plot.
[c]$S_e$ = external surface area from *t*-plot.
[d]$V_t$ = total pore volume determined by BET-plot.
[e]The ZSM-5 prepared without adding seeds was used as the reference sample.

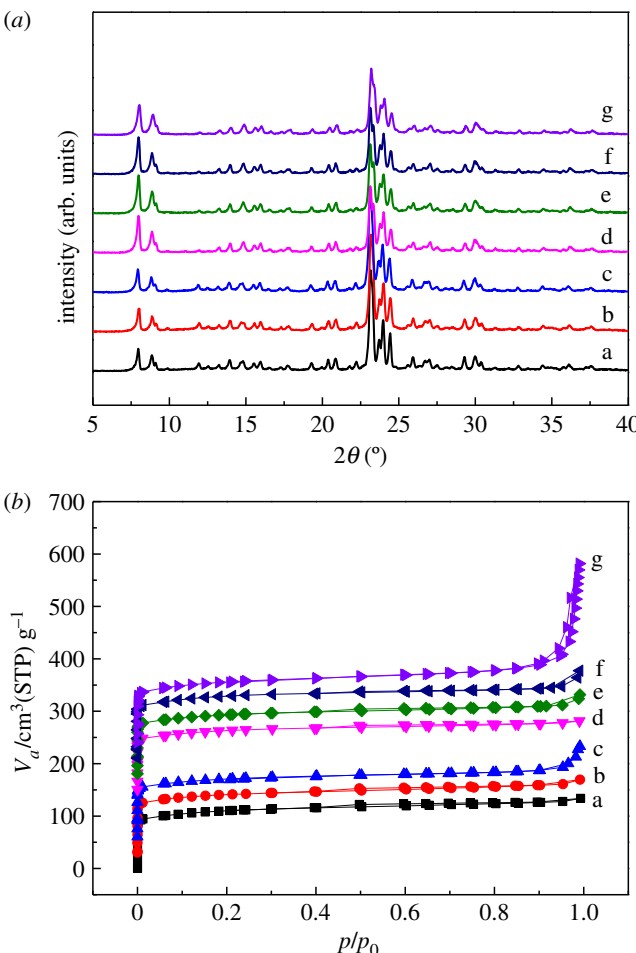

**Figure 6.** XRD patterns (*a*) and N$_2$ adsorption/desorption isotherms (*b*) for ZSM-5 samples with different crystal sizes. a, ZSM-5 (no seeds); b, S80-0.2-1; c, S80-2-0.5; d, S60-0.004-2; e, S60-0.04-1; f, S60-0.2-0.5; and g, S60-2-0.25.

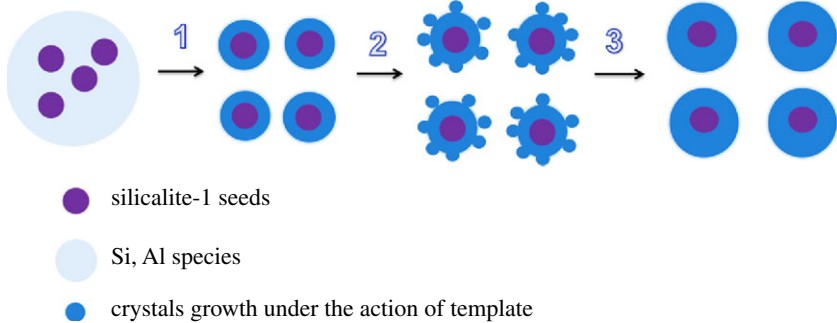

**Figure 7.** The growth mechanism of the ZSM-5 zeolite with adding the silicalite-1 seeds.

the template TPAOH, Si and Al species themselves will also begin to aggregate and grow into some small crystals around the surface. With the further extension of crystallization time, the crystal growth rapidly occurs on the surface of silicalite-1 seeds to form the ZSM-5 crystals of a certain size. The ZSM-5 crystals are successfully synthesized with uniform particle size and core (silicalite-1)–shell (ZSM-5) structure.

## 3.3. Methanol conversion to aromatics reaction

### 3.3.1. Catalytic performance over HZSM-5 zeolites with different crystal sizes

Figure 8 gives the conversion of methanol with time on stream (TOS) over HZSM-5 catalysts with different crystal sizes. It was found that the catalyst lifetime depended on the crystal size and the synthesized

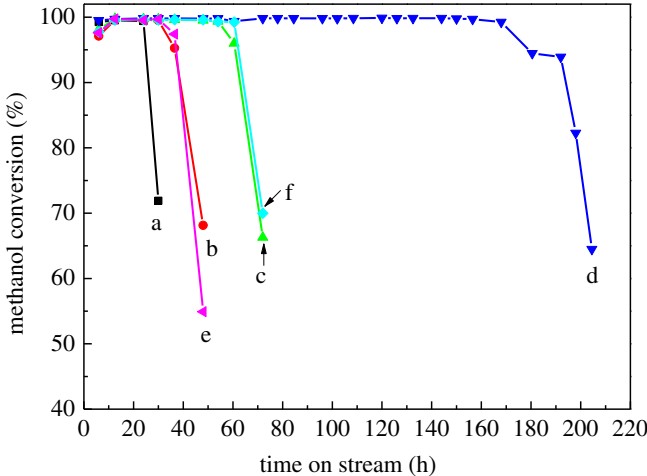

**Figure 8.** Conversion of methanol with time on stream (TOS) over HZSM-5 catalysts of different crystal sizes with the same core and the same crystal size with different cores. a, S60-0.004-2; b, S60-0.04-1; c, S60-0.2-0.5; d, S60-2-0.25; e, S80-0.2-1; and f, S80-2-0.5.

**Table 2.** Product distributions for HZSM-5 zeolites with different crystal sizes.

| catalysts | S60-0.004-2 | S60-0.04-1 | S60-0.2-0.5 | S60-2-0.25 | S80-0.2-1 | S80-2-0.5 |
|---|---|---|---|---|---|---|
| methanol conversion (%) | 99.8 | 99.6 | 99.8 | 99.7 | 99.7 | 99.6 |
| product distribution (wt.%) | | | | | | |
| $C_1$–$C_4$ alkanes | 38.9 | 39.5 | 39.4 | 38.6 | 39.6 | 38.7 |
| $C_2$–$C_5$ alkenes | 8.0 | 6.0 | 4.5 | 4.4 | 6.2 | 4.7 |
| $C_5^+$ hydrocarbons | 51.3 | 53.2 | 54.4 | 55.5 | 52.8 | 55.0 |
| others ($H_2$, $CO_x$) | 1.8 | 1.3 | 1.7 | 1.5 | 1.4 | 1.6 |
| composition of $C_5^+$ hydrocarbons | | | | | | |
| $C_5^+$ nonaromatics | 17.9 | 18.6 | 18.8 | 19.3 | 18.0 | 18.9 |
| BTX | 23.3 | 23.1 | 22.7 | 20.9 | 22.9 | 22.5 |
| $C_9^+$ aromatics | 10.1 | 11.5 | 12.9 | 15.3 | 12.0 | 13.4 |

Data obtained at the time on stream (TOS) of 12.5 h.

HZSM-5 catalyst with a smaller crystal size showed longer catalyst lifetime than that with larger crystal size in the MTA reaction (the catalytic life of S60-2-0.25 can reach over 170 h). With the increase of the crystal size, the catalytic stability of HZSM-5 zeolites decreased significantly. The activity and lifetime of HZSM-5 zeolites with the same particle size and different thickness of silicalite-1 core were almost the same, indicating that the size of the silicalite-1 core had little influence on the reactivity (table 2).

On the other hand, the crystal size of HZSM-5 zeolites had a remarkable influence on the product distribution. Rownaghi *et al*. [19,56] found that the selectivity of aromatics and $C_2$–$C_5$ alkenes on nano-sized ZSM-5 zeolites was greatly improved compared with that on micron ZSM-5 zeolites. As shown in table 2, the small-sized HZSM-5 zeolites showed much higher selectivity to the $C_5^+$ hydrocarbons and aromatics products, while large-sized HZSM-5 zeolites were conducive to the generation of $C_2$–$C_5$ alkenes. The selectivity of $C_5^+$ hydrocarbons increased with decreasing the particle size of HZSM-5 zeolites, whereas the selectivity of $C_2$–$C_5$ alkenes decreased. Meanwhile, the composition of $C_5^+$ hydrocarbons also changed regularly with the particle size. The selectivity of $C_9^+$ aromatics increased with the decrease of crystal size, and high selectivity for BTX was exhibited on the large crystallites. This phenomenon was probably related to the larger external surface area and more pore mouths on small-sized ZSM-5 zeolites, which were responsible for making $C_5^+$ hydrocarbons and aromatics, especially $C_9^+$ aromatics, easily undergo. Furthermore, a larger surface area meant more external surface acid sites, and the light aromatic hydrocarbons (BTX) could continue to take place alkylation or disproportionation on the surface of HZSM-5 that resulted in the improvement of $C_9^+$ aromatics selectivity over the small-sized HZSM-5. Melson & Schüth [57] gravimetrically measured the number

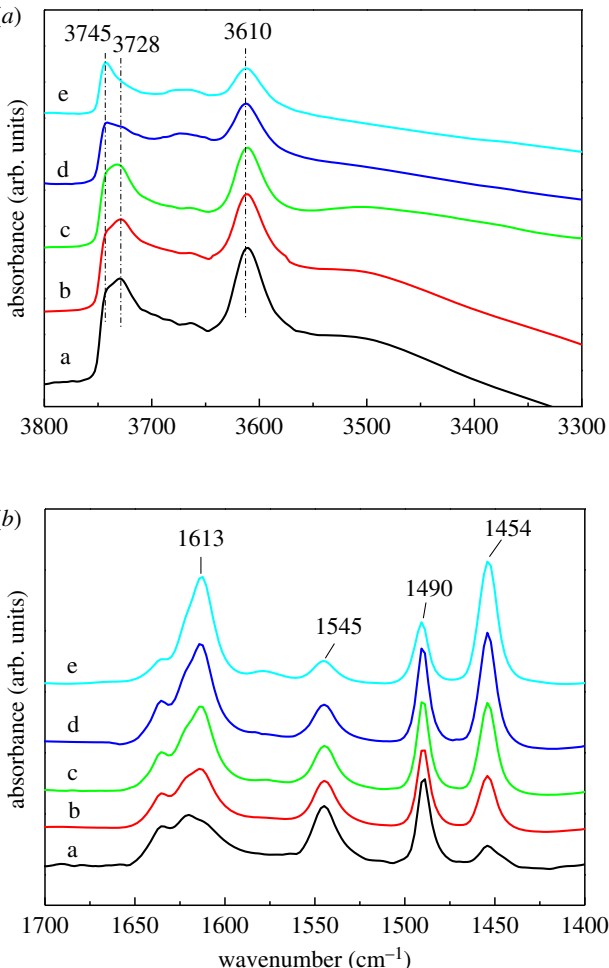

**Figure 9.** FT-IR spectra (*a*) and Py-IR spectra (*b*) of S60-0.2-0.5 and Zn/S60-0.2-0.5: a, S60-0.2-0.5; b, 0.5%Zn/S60-0.2-0.5; c, 1%Zn/S60-0.2-0.5; d, 3%Zn/S60-0.2-0.5; and e, 5%Zn/S60-0.2-0.5.

of external acids of HZSM-5 samples by adsorbing 2,6-dimethylpyridine. The results showed that the amount of external acids increased with decreasing the crystal size of HZSM-5 zeolites. The activity and the shape-selectivity in the disproportionation of ethylbenzene were strongly dependent on the external acidic sites which in turn correlated well with the crystal size.

### 3.3.2. Catalytic performance over Zn/HZSM-5 zeolites

Although the impregnation of Zn species into ZSM-5 could enhance dehydrogenation and promote aromatization, their presence in excess also accelerated the deactivation of the carbon deposition of the catalyst and enhanced the decomposition of methanol to carbon oxides. Therefore, it is necessary to get a correct balance between the two effects, which depends on the content of Zn loading.

Compared with the HZSM-5 zeolite, Zn modified HZSM-5 zeolites showed similar characteristic peaks, indicating that the addition of Zn had little effect on the framework structure of ZSM-5. No new diffraction peaks of ZnO were found on all the XRD patterns, which indicated that zinc species introduced by impregnation were highly dispersed on ZSM-5 zeolites. Meanwhile, the peak strength of Zn modified HZSM-5 zeolites decreased slightly with the increase of Zn loading (electronic supplementary material, figure S4).

Figure 9*a* shows the FT-IR spectra of the HZSM-5 zeolite and Zn-modified HZSM-5 with different Zn contents. The samples exhibited the usual bands at 3750–3700 cm$^{-1}$ and 3610 cm$^{-1}$, which were ascribed to silanol groups and Brønsted acid sites formed by the OH group of Si(OH)Al, respectively [58]. Both external (3745 cm$^{-1}$) and internal (3728 cm$^{-1}$) silanol groups were detected on the S60-0.2-0.5 sample. After loading Zn, the intensity of both the peaks obviously decreased, especially the internal Si-OH peak at 3728 cm$^{-1}$. With the increase of Zn content, the peak of internal Si-OH decreased gradually

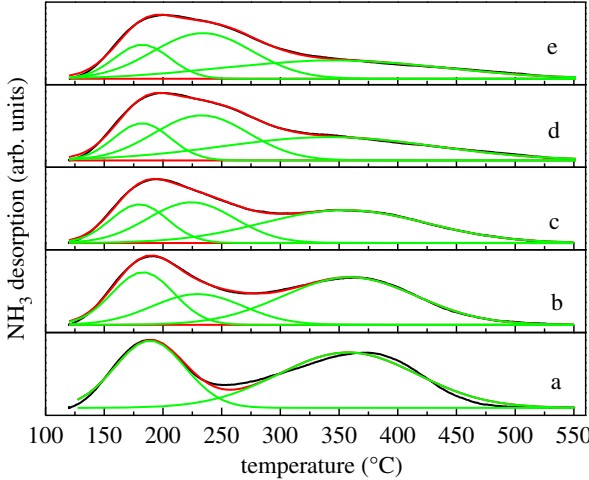

**Figure 10.** NH$_3$-TPD profiles of S60-0.2-0.5 and Zn/S60-0.2-0.5: a, S60-0.2-0.5; b, 0.5%Zn/S60-0.2-0.5; c, 1%Zn/S60-0.2-0.5; d, 3% Zn/S60-0.2-0.5; and e, 5%Zn/S60-0.2-0.5.

**Table 3.** Effect of Zn loading on the surface area, pore volume and acidity of the catalysts. (*S*, strong acids; *M*, medium acids; *W*, weak acids.)

| samples | Zn content (wt.%) ICP | acid strength (mmol g$^{-1}$)[a] | | | acid type (mmol g$^{-1}$) | | $S$ (m$^2$g$^{-1}$)[b] | $V_{mic}$ (cm$^3$ g$^{-1}$)[c] |
|---|---|---|---|---|---|---|---|---|
| | | *W* | *M* | *S* | L acids | B acids | | |
| S60-0.2-0.5 | | 0.20 | | 0.36 | 0.047 | 0.172 | 390 | 0.188 |
| 0.5%Zn/S60-0.2-0.5 | 0.51 | 0.15 | 0.13 | 0.29 | 0.114 | 0.136 | 379 | 0.184 |
| 1%Zn/S60-0.2-0.5 | 1.11 | 0.11 | 0.18 | 0.27 | 0.191 | 0.129 | 362 | 0.172 |
| 3%Zn/S60-0.2-0.5 | 3.05 | 0.10 | 0.25 | 0.22 | 0.231 | 0.110 | 335 | 0.160 |
| 5%Zn/S60-0.2-0.5 | 5.06 | 0.10 | 0.27 | 0.20 | 0.270 | 0.063 | 327 | 0.155 |

[a]Density of the acid sites, assorted according to the acidic strength, determined by NH$_3$-TPD.

[b]Surface area obtained from BET-plot.

[c]$V_{mic}$ = micropore volume determined by *t*-plot.

until it almost disappeared. It indicated that part of the Zn species interacted with Si-OH groups, and the internal Si-OH groups were more likely to interact with Zn species. In addition, the impregnation of Zn also significantly affected the OH groups of Brønsted acid sites. With increasing the Zn content, the IR band at 3610 cm$^{-1}$ decreased gradually, indicating that Zn species interacted with Brønsted acid sites. Moreover, as shown in figure 9*b*, owing to the interaction of zinc with Brønsted acid sites to form ZnOH$^+$ species and generate the new Zn-Lewis acid site, the acid site of Brønsted (1545 cm$^{-1}$) decreased and the acid site of Lewis (1454 cm$^{-1}$) increased after the introduction of Zn [59,60]. Py-IR spectra showed that a new peak appeared at 1613 cm$^{-1}$, which gradually increased with the increase of Zn loading. She *et al*. [61] suggested that the peak at 1613 cm$^{-1}$ was caused by the acid position of the Zn-Lewis acid site. Meanwhile, with the increase of Zn loading, the specific surface area and micropore volume of the catalysts obviously decreased (table 3). Such a result illustrated that Zn species concentrated on the zeolite surface or loaded in the zeolite channels.

NH$_3$-TPD characterization was used to analyse the acid amount and acid strength of the catalysts. As shown in figure 10 and table 3, the introduction of Zn did not change the total acid amount of the zeolite catalysts but evidently changed the distribution of strong and weak acids. Compared with the S60-0.2-0.5 sample without loading Zn, introducing a small amount of Zn (0.5%) reduced the amount of strong acid significantly. Meanwhile, with increasing the Zn content, the amount of strong acid gradually decreased, the low-temperature peak ascribed to the weak acid sites became wider and moved to the high-temperature position. As can be seen, after loading Zn, a peak of medium acids appeared at

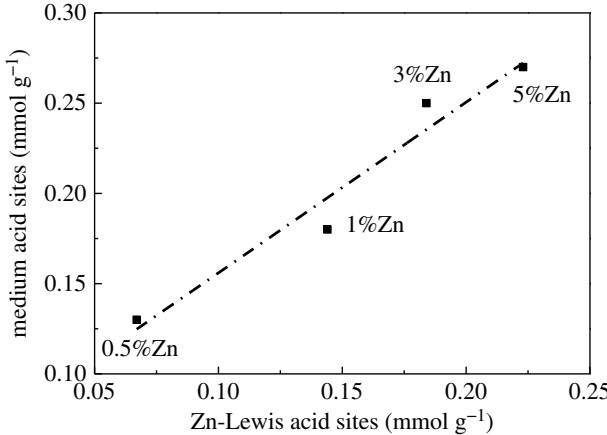

**Figure 11.** Correlation between the amount of medium acid sites and the generated Zn-Lewis acid sites.

230–240°C and the amount of the acids showed an increasing trend with the increase of Zn loading. There was a good linear relationship between the medium acids and Zn-Lewis acids, indicating that the medium acid was attributed to the generation of the Zn-Lewis acids introduced by Zn [62] (figure 11). In addition, the quantity of the acids of the zeolite catalyst was not affected by the Zn content, which also proved that the previously mentioned Zn species interacted with Brønsted acid sites and Si-OH groups should exist in the form of $ZnOH^+$ rather than $O-Zn^{2+}-O$ [44].

Compared with S60-0.2-0.5, Zn/S60-0.2-0.5 impregnated with a small amount of Zn could notably enhance the selectivity of aromatics (figure 12a). With the increase of Zn loading, the selectivity of $C_1$–$C_4$ alkanes and $C_5^+$ aliphatic hydrocarbons decreased gradually, while the selectivity of $C_2$–$C_5$ alkenes increased. The selectivity of aromatics firstly increased and then decreased slowly and reached the maximum value when the Zn content was 1%. The selectivity of aromatics improved from 35.6% to 43.1%. The significant decrease of light alkanes indicated that Zn species could enhance the dehydrogenation ability of light alkanes, and the decrease of Brønsted acid could suppress the hydrogen transfer reaction, thus promote the improvement of aromatics selectivity [37,63]. The increase of $C_2$–$C_5$ alkenes illustrated that the formation of olefins could be obtained either by cracking the reaction on Brønsted acid site or by dehydrogenation reaction on the Zn-Lewis acid site. The variation trend of aromatics selectivity showed that the aromatics selectivity was not only related to the Zn species but also related to the Brønsted acid site. Brønsted acid induced the generation of olefins for methanol transformation, and further, the reactions of dehydrogenation, cyclooligomerization and hydrogen transfer occurred to form aromatic hydrocarbons and light alkanes. Zn-Lewis acid was effective in improving the dehydrogenation of alkanes and promoting alkenes dehydrogenation to aromatic hydrocarbons. The improvement of methanol aromatization was attributed to the synergistic effect of Zn-Lewis acid and Brønsted acid [45]. In addition, Brønsted acid and Zn-Lewis acid also had a certain negative effect on the activity of the MTA reaction. Wang *et al.* [64] found that the selectivity of aromatics was highest when the Zn loading was 1–2%. When the Zn loading was more than 2%, not only did the yield of aromatics decrease but also the cracking reaction of methanol to carbon monoxide and carbon dioxide was enhanced. Brønsted acid sites catalysed the hydrogen transfer reaction and increased the generation of light alkanes, thus reducing the selectivity of aromatic hydrocarbons. Excessive Zn-Lewis acid sites could promote methanol cracking which led to an increase in the products of $H_2$ and $CO_X$.

As shown in figure 12b, the lifetime of the catalyst was reduced gradually with the increase of Zn loading. When the active Zn-Lewis acid sites were generated by the impregnation of Zn, the inactive ZnO clusters were also formed which has been reported in our previous studies [13,44]. These ZnO species dispersed on the surface or in the channels of ZSM-5 zeolites decreased the specific surface area and pore volume. In the MTA reaction, the deactivation of the catalysts was mainly owing to the blockage of the pore mouth or channel caused by carbon deposition which would inhibit the diffusion of the products and accelerate the deactivation of the catalyst [65,66]. It was reported that the large ZnO macroparticles, ZnO nanoparticles and $ZnOH^+$ species were found on Zn/HZSM-5 zeolites after the modification of Zn loading. The micron-sized ZnO had little effect on the physico-chemical properties of the zeolites, while nano-ZnO particles could exist at the pore mouth or inside the channel which was the main reason for the deactivation of Zn modified zeolite catalysts [62]. In order to solve the fast deactivation of catalytic performance of Zn modified ZSM-5 zeolites, the research on the preparation of nano-sized ZSM-5 and mesoporous ZSM-5 zeolites has attracted much

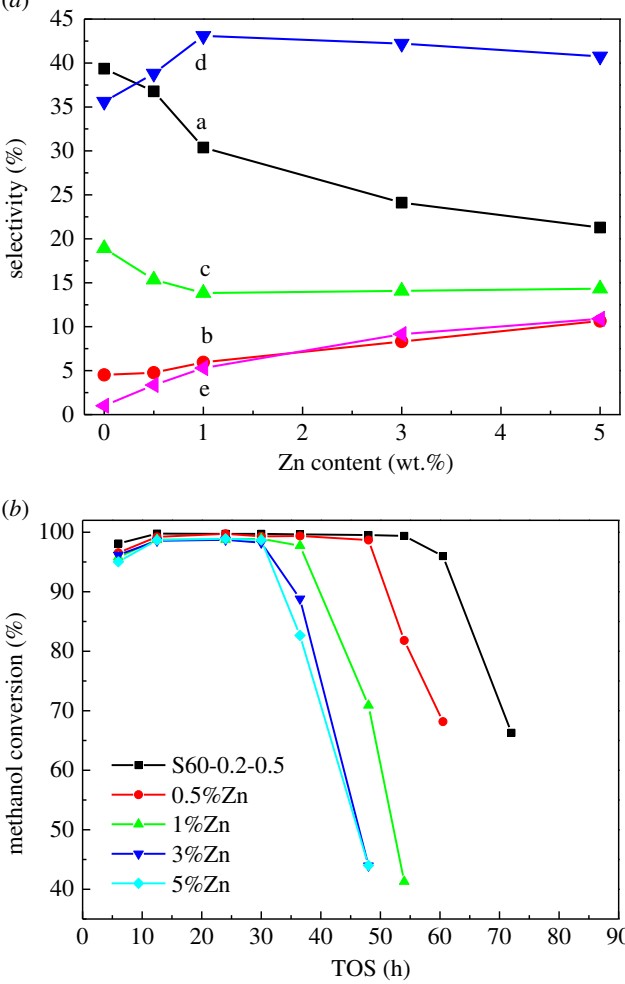

**Figure 12.** (a) Effect of Zn content on product distribution (data obtained at 12.5 h in TOS) and (b) catalyst lifetime. a, $C_1$–$C_4$ alkanes; b, $C_2$–$C_5$ alkenes; c, $C_5^+$-non-aromatic hydrocarbons; d, aromatics; and e, $H_2$ and $CO_X$.

attention. It was found that introducing intracrystalline mesopores into ZSM-5 zeolites significantly improved the stability of the catalysts [45,46,67–69].

## 4. Conclusion

The crystal size of ZSM-5 zeolites synthesized through the hydrothermal synthesis with commercial silica sol could be controlled by adding colloidal silicalite-1 seeds. The ZSM-5 zeolites with controllable crystal size in the range of 200–2200 nm could be obtained by adding the silicalite-1 seeds with different amounts and synthesis conditions. Especially in the scale of 200–1000 nm, the crystal size of ZSM-5 zeolites could be adjusted within 100 nm by varying the amount and size of the seeds. The prepared ZSM-5 zeolites were the composites of silicalite-1@ZSM-5 with the core–shell structure. The crystal size had a significant impact on not only the product distribution but also the stability of the MTA reaction. The catalysts with smaller crystal sizes favoured the production of $C_5^+$ hydrocarbons and $C_9^+$ aromatics, while catalysts with bigger crystal sizes promoted the selectivity of BTX. On the other hand, the catalytic stability of HZSM-5 in the MTA reaction considerably increased with decreased crystal size. The introduction of Zn into ZSM-5 promoted the selectivity for aromatic hydrocarbons in the MTA reaction. The selectivity of aromatics firstly increased and then decreased with the increase of Zn content. The optimal Zn loading was 1%. The improvement of methanol aromatization was attributed to the synergistic effect of Zn-Lewis acid and Brønsted acid.

Data accessibility. Electronic supplementary material (electronic supplementary material, figures S1–S4) is available online at http://doi.org/10.5061/dryad.t1g1jwt2x [70].

Authors' contributions. X.N.: conceptualization, data curation, methodology, writing—original draft, writing—review and editing; Y.B.: data curation, formal analysis, writing—review and editing; Y.-e.D.: data curation, formal analysis, writing—review and editing; H.Q.: data curation, investigation; Y.C.: conceptualization, supervision, writing—original draft.

All authors gave final approval for publication and agreed to be held accountable for the work performed therein.

Competing interests. The authors declare no competing interests.

Funding. This work was financially supported by the Scientific and Technological Innovation Programs of Higher Education Institutions in Shanxi (grant nos. 2019L0881 and 2020L0612), Doctor Research Funds of Jinzhong University, the Applied Basic Research Project of Shanxi (grant nos. 201901D111303 and 201901D111299), Shanxi '1331 Project' Key Innovative Research Team (grant no. PY201817) and Jinzhong University '1331 Project' Key Innovative Research Team (grant no. jzxycxtd2019005).

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
