## [Peer Review File · Royal Society Open Science]

Review History

RSOS-211284.R0 (Original submission)

Review form: Reviewer 1

Is the manuscript scientifically sound in its present form?

Yes

Are the interpretations and conclusions justified by the results?

Yes

Is the language acceptable?

Yes

Do you have any ethical concerns with this paper?

No

Have you any concerns about statistical analyses in this paper?

No

Recommendation?

Accept with minor revision (please list in comments)

Comments to the Author(s)

The manuscript reported that the Size Controllable Synthesis of ZSM-5 Zeolite and its Catalytic Performance in the Reaction of Methanol Conversion to Aromatics. It is very interesting, some question should be addressed before accept.

1. In Figure 1, what is meaning for S60 and S80 ?
2. In Figure 3, the (a) S60-0.004-2, (b) S60-0.04-1, (c) S60-0.2-0.5, (d) S60-2-0.25, (e) S80-0.2-1 and (f) S80-2-0.5 were represented some samples, please clarify it.
3. From figure 2, when the crystallization time is longer, the crystal size of ZSM-5 is small, what is reason?
4. Some references is relevant to the manuscript, please cited it, such as: Direct conversion of cellulose to levulinic acid using SO₃H-functionalized ionic liquids containing halogen-anions[J]. Journal of Molecular Liquids. 2021, 339, 117278.

Review form: Reviewer 2**Is the manuscript scientifically sound in its present form?**

Yes

Are the interpretations and conclusions justified by the results?

No

Is the language acceptable?

Yes

Do you have any ethical concerns with this paper?

No

Have you any concerns about statistical analyses in this paper?

No

Recommendation?

Major revision is needed (please make suggestions in comments)

Comments to the Author(s)

In this work, Niu and coworkers presented an investigation on the synthesis and performance of ZSM-5 zeolite in methanol to aromatics conversion (MTA). They also investigated the impact of Zn loading on ZSM-5 to the product distribution in MTA. By carefully reading, I found the following should be addressed before the manuscript can be considered further.

There are many reported works on seeded synthesis of zeolite as well as the MTO process and the impact of Zn to the product distribution. It would be nice if relevant results were cited and compared together.

In Figure 2, the authors are suggested to do a statistics investigation to see the change of size of particles of ZSM-5 samples with the crystallization time. While discussing Figure 2, the authors mentioned fine tuning of ZSM-5 particle size by seeded crystallization. Though Figure 3 is related to the impact of number seed particles used on the particle size distribution, the particle size is already beyond 200 nm. The discussion should be expended as the current discussion and results don't support the fine tuning feature.

- In Figure 4, the authors are suggested to include data on other synthesis conditions, such as those shown in Figure 2 and Figure 3. Is there any correlation between the particle size and the crystallinity? The definition and method to characterize crystallinity should be mentioned briefly in the context.
- While discussing the performance of HZSM-5 in MTA in Table 5, the correlations among the conversion of methanol, C5+ hydrocarbons and composition of C5+ hydrocarbons with the particle size is not clear. Why?
- For Figure 11, how is the water content in the products? The factors that account for the fast decay of the catalytic performance of the Zn modified ZSM-5 should be discussed.
- Discussion on potential approaches to address the fast decay of catalytic performance of Zn modified ZSM-5 should be mentioned at least with cited references.

Decision letter (RSOS-211284.R0)

Dear Dr Niu:

Title: Size Controllable Synthesis of ZSM-5 Zeolite and its Catalytic Performance in the Reaction of Methanol Conversion to Aromatics
Manuscript ID: RSOS-211284

The editor assigned to your manuscript has now received comments from reviewers. We would like you to revise your paper in accordance with the referee and Subject Editor suggestions which can be found below (not including confidential reports to the Editor). Please note this decision does not guarantee eventual acceptance.

Please submit your revised paper before 28-Jan-2022. Please note that the revision deadline will expire at 00.00am on this date. If we do not hear from you within this time then it will be assumed that the paper has been withdrawn. In exceptional circumstances, extensions may be possible if agreed with the Editorial Office in advance. We do not allow multiple rounds of revision so we urge you to make every effort to fully address all of the comments at this stage. If deemed necessary by the Editors, your manuscript will be sent back to one or more of the original reviewers for assessment. If the original reviewers are not available we may invite new reviewers.

Please also include the following statements alongside the other end statements. As we cannot publish your manuscript without these end statements included, if you feel that a given heading is not relevant to your paper, please nevertheless include the heading and explicitly state that it is not relevant to your work.

- Ethics statement

Please clarify whether you received ethical approval from a local ethics committee to carry out your study. If so please include details of this, including the name of the committee that gave consent in a Research Ethics section after your main text. Please also clarify whether you received informed consent for the participants to participate in the study and state this in your Research Ethics section.

OR

Please clarify whether you obtained the necessary licences and approvals from your institutional animal ethics committee before conducting your research. Please provide details of these licences and approvals in an Animal Ethics section after your main text.

OR

Please clarify whether you obtained the appropriate permissions and licences to conduct the fieldwork detailed in your study. Please provide details of these in your methods section.

- Data accessibility

It is a condition of publication that you make available the data and research materials supporting the results in the article. Datasets should be deposited in an appropriate publicly available repository and details of the associated accession number, link or DOI to the datasets must be included in the Data Accessibility section of the article (<https://royalsocietypublishing.org/rsos/for-authors#question17>). Reference(s) to datasets should also be included in the reference list of the article with DOIs (where available).

Please include a Data Availability section after your main text stating where supporting data are available from, or where they will be made available should your article be accepted for publication.

If you wish to submit your supporting data or code to Dryad (<http://datadryad.org/>), or modify your current submission to dryad, please use the following link:
<http://datadryad.org/submit?journalID=RSOS&manu=RSOS-211284>

- Competing interests

Please include a Competing Interests section after your main text declaring any financial or non-financial competing interests. If you have no competing interests please state 'I/we have no competing interests.'

- Authors' contributions

Please include an Authors' Contributions section at the end of your main text detailing the contribution of each author. All authors should have read and approved the manuscript before submission and this should be stated in the Authors' Contributions section.

The list of Authors should meet all of the following criteria; 1) substantial contributions to conception and design, or acquisition of data, or analysis and interpretation of data; 2) drafting the article or revising it critically for important intellectual content; and 3) final approval of the version to be published.

• Acknowledgements

• Funding statement

Please include a funding section after your main text which lists the source of funding for each author.

Yours sincerely,
Dr Ellis Wilde
Publishing Editor, Journals

On behalf of the Subject Editor Professor Anthony Stace and the Associate Editor Dr Annette Trunschke.

RSC Associate Editor
Comments to the Author:
(There are no comments.)

RSC Subject Editor
Comments to the Author:
(There are no comments.)

Reviewers' Comments to Author:

Reviewer: 1

Comments to the Author(s)

The manuscript reported that the Size Controllable Synthesis of ZSM-5 Zeolite and its Catalytic Performance in the Reaction of Methanol Conversion to Aromatics. It is very interesting, some question should be addressed before accept.

1. In Figure 1, what is meaning for S60 and S80 ?

2. In Figure 3, the (a) S60-0.004-2, (b) S60-0.04-1, (c) S60-0.2-0.5, (d) S60-2-0.25, (e) S80-0.2-1 and (f) S80-2-0.5 were represented some samples, please clarify it.
3. From figure 2, when the crystallization time is longer, the crystal size of ZSM-5 is small, what is reason?
4. Some references is relevant to the manuscript, please cited it, such as: Direct conversion of cellulose to levulinic acid using SO₃H-functionalized ionic liquids containing halogen-anions[J]. Journal of Molecular Liquids. 2021, 339, 117278.

Reviewer: 2

Comments to the Author(s)

In this work, Niu and coworkers presented an investigation on the synthesis and performance of ZSM-5 zeolite in methanol to aromatics conversion (MTA). They also investigated the impact of Zn loading on ZSM-5 to the product distribution in MTA. By carefully reading, I found the following should be addressed before the manuscript can be considered further.

There are many reported works on seeded synthesis of zeolite as well as the MTO process and the impact of Zn to the product distribution. It would be nice if relevant results were cited and compared together.

In Figure 2, the authors are suggested to do a statistics investigation to see the change of size of particles of ZSM-5 samples with the crystallization time. While discussing Figure 2, the authors mentioned fine tuning of ZSM-5 particle size by seeded crystallization. Though Figure 3 is related to the impact of number seed particles used on the particle size distribution, the particle size is already beyond 200 nm. The discussion should be expanded as the current discussion and results don't support the fine tuning feature.

In Figure 4, the authors are suggested to include data on other synthesis conditions, such as those shown in Figure 2 and Figure 3. Is there any correlation between the particle size and the crystallinity? The definition and method to characterize crystallinity should be mentioned briefly in the context.

While discussing the performance of HZSM-5 in MTA in Table 5, the correlations among the conversion of methanol, C₅+ hydrocarbons and composition of C₅+ hydrocarbons with the particle size is not clear. Why?

For Figure 11, how is the water content in the products? The factors that account for the fast decay of the catalytic performance of the Zn modified ZSM-5 should be discussed.

Discussion on potential approaches to address the fast decay of catalytic performance of Zn modified ZSM-5 should be mentioned at least with cited references.

Author's Response to Decision Letter for (RSOS-211284.R0)

See Appendix A.

RSOS-211284.R1 (Revision)

Review form: Reviewer 1

Is the manuscript scientifically sound in its present form?

Yes

Are the interpretations and conclusions justified by the results?

Yes

Is the language acceptable?

Yes

Do you have any ethical concerns with this paper?

Yes

Have you any concerns about statistical analyses in this paper?

No

Recommendation?

Accept as is

Comments to the Author(s)

The manuscript is well revised

Review form: Reviewer 2

Is the manuscript scientifically sound in its present form?

Yes

Are the interpretations and conclusions justified by the results?

Yes

Is the language acceptable?

Yes

Do you have any ethical concerns with this paper?

No

Have you any concerns about statistical analyses in this paper?

No

Recommendation?

Accept as is

Comments to the Author(s)

As the authors have addressed my concerns, I think it publishable now.

Decision letter (RSOS-211284.R1)

Dear Dr Niu:

Title: Size Controllable Synthesis of ZSM-5 Zeolite and its Catalytic Performance in the Reaction of Methanol Conversion to Aromatics
Manuscript ID: RSOS-211284.R1

It is a pleasure to accept your manuscript in its current form for publication in Royal Society Open Science. The chemistry content of Royal Society Open Science is published in collaboration with the Royal Society of Chemistry.

Yours sincerely,
Ellis Wilde
Publishing Editor, Journals

On behalf of the Subject Editor Professor Anthony Stace and the Associate Editor Dr Annette Trunschke.

RSC Associate Editor
Comments to the Author:
(There are no comments.)

RSC Subject Editor
Comments to the Author:
(There are no comments.)

Reviewer(s)' Comments to Author:
Reviewer: 2
Comments to the Author(s)
As the authors have addressed my concerns, I think it publishable now.

Reviewer: 1
Comments to the Author(s)
The manuscript is well revised

Appendix A

Dear Dr Ellis Wilde,

The manuscript with the number of *RSOS-211284* entitled

“Size Controllable Synthesis of ZSM-5 Zeolite and its Catalytic Performance in the Reaction of Methanol Conversion to Aromatics”

by Xianjun Niu, Yang Bai, Yi-en Du, Hongxue Qi and Yongqiang Chen has been revised with the considerations of reviewers’ comments and is ready for your consideration for publication in *Royal Society Open Science*.

Thanks to the editors and reviewers for the informative and constructive advices and kind efforts in handling this manuscript. You will see that the manuscript is improved greatly with the help of these advices. I hope you would be contented with our revision work. We would be very grateful if the manuscript could be soon accepted for publication.

The revised version and responses to reviewers were uploaded in the system. All the revisions were highlighted in red (revised according to the reviewers).

Thanks for your kind efforts.

Best regards

Sincerely yours,

Xianjun Niu

Response to Reviewer 1 Comments

Point 0: The manuscript reported that the Size Controllable Synthesis of ZSM-5 Zeolite and its Catalytic Performance in the Reaction of Methanol Conversion to Aromatics. It is very interesting, some question should be addressed before accept.

Response: Thanks to the reviewer for the positive and informative comments and instructive revision advices. The manuscript was revised greatly by considering the reviewer's opinion. All the revisions were highlighted in red.

Point 1: In Figure 1, what is meaning for S60 and S80?

Response 1: Thanks to the reviewer for pointing out this issue and giving the valuable advices. The silicalite-1 seeds synthesized at 60 °C for 20 days and 80 °C for 6 days were denoted as S60 and S80, respectively. In order to facilitate the readers to better understand, we have made a more detailed explanation in the experimental section. S60 and S80 are also annotated in the caption of Figure 1. We correct it in the revised manuscript and the corresponding revision is in section 2.1 and the caption of Figure 1. We paste these here for your check:

“The silicalite-1 seeds synthesized at 60 °C and 80 °C were denoted as S60 and S80, respectively. Without specifying the crystallization time of the seeds, S60 and S80 respectively represented the seeds synthesized at 60 °C for 20 days and 80 °C for 6 days.” (Section 2.1, Page 3)

“Figure 1. TEM images of silicalite-1seeds. (a) S60 - Seeds synthesized at 60 °C for 20 days and (b) S80 - Seeds synthesized at 80 °C for 6 days.” (The caption of Figure 1, Page 6)

Point 2: In Figure 3, the (a) S60-0.004-2, (b) S60-0.04-1, (c) S60-0.2-0.5, (d)S60-2-0.25, (e) S80-0.2-1and (f) S80-2-0.5 were represented some samples, please clarify it.

Response 2: Thanks to the reviewer for pointing out this issue. In order to display the effect of seed content and seed size on the particle size of the synthesized ZSM-5 zeolites more directly, the synthesized ZSM-5 samples were symbolized. We explained the symbol abbreviations of samples at the end of the first paragraph of 2.1 part.

“The obtained samples were designated as S_x-y-z, in which S stood for the silicalite-1 seed used in the synthesis of ZSM-5, x represented the synthesis temperature (in °C) of the silicalite-1 seed, y and z represented the weight percent of the seed and the particle diameter of the synthesized ZSM-5 sample in micrometers, respectively.” (Section 2.1, Page 4)

Point 3: From figure 2, when the crystallization time is longer, the crystal size of ZSM-5 is small, what is reason?

Response 3: Thanks to the reviewer for pointing out this issue. In figure 2, the effect of prolongation of the *seed* crystallization time on the particle size of the prepared ZSM-5 zeolites was investigated. The prepared colloidal silicalite-1 crystalline mixture that had not undergone any process treatment was used as the seeds to synthesize ZSM-5 zeolite. Therefore, when adding the same amount of the untreated colloidal seed into the synthesis system of ZSM-5 zeolite, more and more silicalite-1 crystal nuclei were formed as the crystallization time of the seed was prolonged. The increase of seed crystal nuclei resulted in the decrease of particle size of the synthesized ZSM-5. In the synthesis of ZSM-5, we added untreated colloidal seed rather than treated solid seed. In order to facilitate the readers to better understand, we have made a more detailed explanation in the experimental section of 2.1 and the discussion section of figure 2. We correct it in the revised manuscript and the corresponding revision is in the experimental section of 2.1 and the discussion section of figure 2 marked in red. We paste these here for your check:

“The prepared colloidal silicalite-1 crystalline mixture that had not undergone any process treatment was directly used as the seeds to synthesize ZSM-5 zeolite.” (Section 2.1, Page 4)

“Therefore, when adding the same amount of the colloidal seeds without any centrifugation or washing process into the synthesis system of ZSM-5 zeolite, silicalite-1 seeds prepared within 10 days had relatively few crystals nucleus and nonuniform particle size, leading to relatively larger particle size and nonuniform in size and distribution of the synthesized ZSM-5 zeolite. More and more silicalite-1 crystal nuclei in the gel solution were formed as the crystallization time of the seed was prolonged. The increase of seed crystal nuclei resulted in the decrease of particle size of the synthesized ZSM-5 zeolite.” (The discussion section of figure 2, Page 7)

Point 4: Some references is relevant to the manuscript, please cited it, such as: Direct conversion of cellulose to levulinic acid using SO₃H-functionalized ionic liquids containing halogen-anions[J]. *Journal of Molecular Liquids*. 2021, 339, 117278.

Response 4: Thanks to the reviewer for pointing out this issue and giving the valuable advices. With considering the reviewer’s comments, some recent related references (Ref. 7,8 and19) about the renewable resource that can serve as an alternative resource for the production of chemicals were added in the introduction section and the references were updated and rearranged. The corresponding revision were shown here:

“Therefore, it is crucial to develop new processes as an alternative to the synthetic route to produce aromatics or other chemicals. [1-8]” (Page 2, Paragraph 1)

“Compared with the large crystallites, the small-sized ZSM-5 crystallites show a long catalyst lifetime and high selectivity of C₅⁺ hydrocarbons and aromatics in MTH reaction. [17-19]” (Page 2, Paragraph 2)

[7] Shi, S.B.; Wu, Y.F.; Zhang, M.T.; Zhang, Z.Q.; Oderinde, O.; Gao, L.J.; Xiao, G.M. Direct conversion of cellulose to levulinic acid using SO₃H-functionalized ionic liquids containing halogen-anions. *J. Mol. Liq.* **2021**, 339, 117278.

[8] Shi, S.B.; Wu, Y.F.; Zhang, M.T.; Wei, R.P.; Gao, L.J.; Xiao, G.M. Multiple-SO₃H functionalized ionic liquid as efficient catalyst for direct conversion of carbohydrate biomass into levulinic acid. *Mol. Catal.* **2021**, 509, 111659.

[19] Rownaghi, A.A.; Hedlund, J. Methanol to Gasoline-Range Hydrocarbons: Influence of Nanocrystal Size and Mesoporosity on Catalytic Performance and Product Distribution of ZSM-5. *Ind. Eng. Chem. Res.* **2011**, *50*, 11872-11878.

Response to Reviewer 2 Comments

Point 0: In this work, Niu and coworkers presented an investigation on the synthesis and performance of ZSM-5 zeolite in methanol to aromatics conversion (MTA). They also investigated the impact of Zn loading on ZSM-5 to the product distribution in MTA. By carefully reading, I found the following should be addressed before the manuscript can be considered further.

Response: Thanks to the reviewer for the positive and informative comments and instructive revision advices. The manuscript was revised greatly by considering the reviewer's opinion. All the revisions were highlighted in red.

Point 1: There are many reported works on seeded synthesis of zeolite as well as the MTO process and the impact of Zn to the product distribution. It would be nice if relevant results were cited and compared together.

Response 1: Thanks to the reviewer for pointing out this issue and giving the valuable advices. With considering the reviewer's comments, the relevant results were cited and compared together. We copy these here for your check:

“It was reported that the particle size of ZSM-5 zeolite can be well controlled in nanometer to submicron size range by seed-induction method. Majano et al. synthesized nanosized ZSM-5 from organic-template-free gel systems containing silicalite-1 seeds. The size of the ZSM-5 crystals prepared at different temperatures varied from 70 nm to 700 nm. [54] Tang and co-workers successfully synthesized ZSM-5 zeolites with adjustable submicron-crystal size (270-1100 nm) by seed-induction method. The addition of crystal seed was beneficial to the formation of ZSM-5 zeolite with smaller particle size because it can accelerate nucleation and inhibit crystal growth. [28,29]” (Last sentence on Page 8, and Page 9, Paragraph 1)

“The change of the particle size of ZSM-5 samples with the crystallization time was investigated. In the initial stage, the gel used to synthesize ZSM-5 was almost full of irregular granules with the crystal size of several tenths nanometers which were smaller than the added S80 seeds (Figure 4a). It was due to slight or partial dissolution of the silicalite-1 seeds when added to the alkaline synthetic gel. [55] In organic-template-free gel system, it was also found that the addition of the seeds into synthetic mixture caused deposition of small amorphous nanoparticles formed by depletion of the silicalite-1 gel. [29]” (Page 9, Paragraph 2)

“On the other hand, the crystal size of HZSM-5 zeolites had a remarkable influence on the product distribution. Rownaghi et al. found that the selectivity of aromatics and C₂-C₅ alkenes on nano-sized ZSM-5 zeolites was greatly improved compared with that on micron ZSM-5 zeolites. [19, 56]” (Page 15, Paragraph 2)

“In addition, Brønsted acid and Zn-Lewis acid also had a certain negative effect on the activity of the MTA reaction. Wang and co-workers found that the selectivity of aromatics was highest when the Zn loading was 1-2%. When the Zn loading was more than 2%, not

only did the yield of aromatics decreased, but also the cracking reaction of methanol to carbon monoxide and carbon dioxide was enhanced. [64]” (Page 21, Paragraph 1)

“In MTA reaction, the deactivation of the catalysts was mainly due to the blockage of pore mouth or channel caused by carbon deposition which would inhibit the diffusion of the products and accelerate the deactivation of the catalyst. [65,66] It was reported that the large ZnO macroparticles, ZnO nanoparticles, and ZnOH⁺ species were found on Zn/HZSM-5 zeolites after the modification of Zn loading. The micron-sized ZnO had little effect on the physicochemical properties of the zeolites, while nano ZnO particles could exist at the pore mouth or inside the channel which was the main reason for the deactivation of Zn modified zeolite catalysts. [61] In order to solve the fast deactivation of catalytic performance of Zn modified ZSM-5 zeolites, the research on the preparation of nano-sized ZSM-5 and mesoporous ZSM-5 zeolites had attracted much attention. It was found that the introducing intracrystalline mesopores into ZSM-5 zeolites significantly improved the stability of the catalysts. [45,46,67-69]” (Page 21, Paragraph 2)

Point 2: In Figure 2, the authors are suggested to do a statistics investigation to see the change of size of particles of ZSM-5 samples with the crystallization time. While discussing Figure 2, the authors mentioned fine tuning of ZSM-5 particle size by seeded crystallization. Though Figure 3 is related to the impact of number seed particles used on the particle size distribution, the particle size is already beyond 200 nm. The discussion should be expanded as the current discussion and results don't support the fine tuning feature.

Response 2: Thanks to the reviewer for pointing out this issue and giving the valuable advices. With considering the reviewer's comments, the change of size of particles of ZSM-5 samples with the crystallization time is investigated (Figure 4 is added.). The inaccurate statement that fine tuning of ZSM-5 particle size by seeded crystallization is corrected and the discussion is expanded. We paste these here for your check:

Figure 4. SEM images of ZSM-5 synthesized with different crystallization times by using 0.8 wt.% S80. (a) 1 h, (b) 2 h, (c) 2.5 h, (d) 5 d, (e) 12 h and (f) 48 h.

“The change of the particle size of ZSM-5 samples with the crystallization time was investigated. In the initial stage, the gel used to synthesize ZSM-5 was almost full of irregular granules with the crystal size of several tenths nanometers which were smaller than the added S80 seeds (Figure 4a). It was due to slight or partial dissolution of the silicalite-1 seeds when

added to the alkaline synthetic gel. [55] In organic-template-free gel system, it was also found that the addition of the seeds into synthetic mixture caused deposition of small amorphous nanoparticles formed by depletion of the silicalite-1 gel. [29] With increasing the crystallization time, the amorphous silica and alumina species located around the seeds started to deposit onto the seeds surface and agglomerate together to generate ZSM-5 crystals with fluffy round morphology (Figure 4b). When the crystallization time reached 2.5h, the final crystals of ZSM-5 were obtained (Figure 4c). The size and morphology of ZSM-5 remained unchanged when the crystallization time was prolonged (Figure 4e-f).” (Page 9, Paragraph 2)

“The ZSM-5 zeolite with uniform and tuneable crystal size especially in the scale of nanometer to submicron can be achieved by adding the seed crystals to the synthesis system. [29]” (Page 7, Paragraph 1)

“The crystal size of ZSM-5 can be tuned from 200 nm to 2200 nm through adjusting the amount of the seeds (0.004-10 wt.%) as well as the synthesis conditions of the seeds (crystallization temperature and time). (Figure S3) Especially in the scale of 200 nm to 1000 nm, the crystal size of ZSM-5 can be adjusted within 100 nm by varying the amount and size of the seeds. The maximum uniform particle size of synthesized ZSM-5 was about 2.2 μm with the addition of the seed crystals. It was reported that the particle size of ZSM-5 zeolite can be well controlled in nanometre to submicron size range by seed-induction method. Majano et al. synthesized nanosized ZSM-5 from organic-template-free gel systems containing silicalite-1 seeds. The size of the ZSM-5 crystals prepared at different temperatures varied from 70 nm to 700 nm. [54] Tang and co-workers successfully synthesized ZSM-5 zeolites with adjustable submicron-crystal size (270-1100 nm) by seed-induction method.” (Page 8, Paragraph 1)

Figure S3. The relationship between the seed amount and the crystal size of the synthesized ZSM-5 zeolites.

Point 3: In Figure 4, the authors are suggested to include data on other synthesis conditions, such as those shown in Figure 2 and Figure 3. Is there any correlation between the particle

size and the crystallinity? The definition and method to characterize crystallinity should be mentioned briefly in the context.

Response 3: Thanks to the reviewer for pointing out this issue and giving the valuable advices.

In this work, we don't find a regular relationship between the particle size and the crystallinity (Table 1), and the relevant research results are rarely reported. Crystallinity represents the degree of crystallization perfection of zeolites and a reference sample is usually selected to compare the degree of crystallization with relative crystallinity. In Figure 4 (Figure 5 in revision), the relative crystallinity values were obtained from XRD data. The relative crystallinity is determined by comparing the total peak areas in the range of $2\theta = 22-25^\circ$ with those of the ZSM-5 zeolite prepared without adding seeds.

With considering the reviewer's comments, we have added the XRD results and corresponding discussions to support figure 4. The definition and method to characterize crystallinity is mentioned briefly in the context. We paste these here for your check:

“Crystallinity represents the degree of crystallization perfection of zeolites and a reference sample is usually selected to compare the degree of crystallization with relative crystallinity. In this work, the relative crystallinity was determined by comparing the total peak areas in the range of $2\theta = 22-25^\circ$ with those of the ZSM-5 zeolite prepared without adding seeds.” (Page 4, Paragraph 3)

Figure 5. XRD patterns of ZSM-5 synthesized with different crystallization times by using 0.8 wt.% S80 (A)

“Figure 5A gave the XRD patterns of the ZSM-5 zeolites synthesized with different crystallization times by using 0.8 wt.% S80. At the initial stage of crystallization (1h), only partially dissolved seeds with damaged structures existed. Therefore, the peak strength of XRD was very weak. With prolonging the crystallization time, ZSM-5 crystals began to form and the peak strength increased obviously. Finally, the perfect ZSM-5 crystals were prepared after crystallization for 2.5 h and the peak intensity hardly changed at all, consistent with

above SEM results. The relative crystallinity was determined by comparing the total peak areas in the range of $2\theta = 22-25^\circ$ with those of the ZSM-5 zeolite prepared without adding the seeds. The decrease of the particle size had little effect on the crystallinity of ZSM-5 zeolites (Table 1).” (Page 10, Paragraph 1)

Point 4: While discussing the performance of HZSM-5 in MTA in Table 2, the correlations among the conversion of methanol, C₅⁺ hydrocarbons and composition of C₅⁺ hydrocarbons with the particle size is not clear. Why?

Response 4: Thanks to the reviewer for pointing out this issue.

The conversion of methanol over all catalysts is greater than 99.5%, which could be considered as complete conversion of methanol. As shown in table 2, the data in the first four columns and the last two columns are the results of ZSM-5 zeolites synthesized by different seeds. The particle sizes of the samples in the first four columns (2, 1, 0.5 and 0.25 μ m) and the last two columns (1 and 0.5 μ m) gradually decreased, respectively. The C₅⁺ hydrocarbons and composition of C₅⁺ hydrocarbons (C₅⁺- nonaromatics, BTX and C₉⁺ aromatics) all changed regularly with the particle size of ZSM-5 zeolites. The selectivity of C₅⁺ hydrocarbons increased with decreasing the particle size of HZSM-5 zeolites and the composition of C₅⁺ hydrocarbons also changed regularly with the particle size. The selectivity of C₉⁺ aromatics and C₅⁺ nonaromatics in C₅⁺ hydrocarbons increased with the decrease of crystal size, whereas the selectivity of BTX in C₅⁺ hydrocarbons decreased.

Table 2. Product distributions for HZSM-5 zeolites with different crystal sizes.

Catalysts	S60-0.004-2	S60-0.4-1	S60-0.2-0.5	S60-2-0.25	S80-0.2-1	S80-2-0.5
Methanol conversion (%)	99.8	99.6	99.8	99.7	99.7	99.6
Product distribution (wt.%)						
C ₁ -C ₄ alkanes	38.9	39.5	39.4	38.6	39.6	38.7
C ₂ -C ₅ alkenes	8.0	6.0	4.5	4.4	6.2	4.7
C ₅ ⁺ hydrocarbons	51.3	53.2	54.4	55.5	52.8	55.0
Others (H ₂ , CO _x)	1.8	1.3	1.7	1.5	1.4	1.6
Composition of C ₅ ⁺ hydrocarbons						
C ₅ ⁺ nonaromatics	17.9	18.6	18.8	19.3	18.0	18.9
BTX	23.3	23.1	22.7	20.9	22.9	22.5
C ₉ ⁺ aromatics	10.1	11.5	12.9	15.3	12.0	13.4

With considering the reviewer's comments, in order to more clearly reflect the correlations among C₅⁺ hydrocarbons and composition of C₅⁺ hydrocarbons with the particle size, we rewrite the discussion results of this part. We paste these here for your check:

“On the other hand, the crystal size of HZSM-5 zeolites had a remarkable influence on the product distribution. Rownaghi et al. found that the selectivity of aromatics and C₂-C₅ alkenes on nano-sized ZSM-5 zeolites was greatly improved compared with that on micron ZSM-5 zeolites. [19, 56] As shown in table 2, the small-sized HZSM-5 zeolites showed much higher selectivity to the C₅⁺ hydrocarbons and aromatics products, while large-sized HZSM-5 zeolites were conducive to the generation of C₂-C₅ alkenes. The selectivity of C₅⁺ hydrocarbons increased with decreasing the particle size of HZSM-5 zeolites, whereas the selectivity of C₂-C₅ alkenes decreased. Meanwhile, the composition of C₅⁺ hydrocarbons also changed regularly with the particle size.” (Page 15, Paragraph 2)

Point 5: For Figure 11, how is the water content in the products? The factors that account for the fast decay of the catalytic performance of the Zn modified ZSM-5 should be discussed.

Response 5: Thanks to the reviewer for pointing out this issue and giving the valuable advices.

In the reaction of methanol conversion to hydrocarbons, product distribution generally refers to the distribution of different types of hydrocarbons in the total hydrocarbons. In MTA reaction, there is only methanol in the feed. The product water comes from H and O atoms in methanol. The theoretical yield of water is about 56% when methanol is completely converted to hydrocarbons. In figure 11(Figure 12 in revision), the water content in the products is 55.39% (S60-0.2-0.5), 53.31% (0.5%Zn), 53.02% (1%Zn), 52.67% (3%Zn) and 52.32%% (5%Zn), respectively. After the introduction of Zn, the water content decreased due to the promotion of methanol cracking. The results are consistent with the changes of CO_x in the figure.

With considering the reviewer's comments, the calculation instructions of methanol conversion and product selectivity are added in the end of experimental part, and the factors that account for the fast decay of the catalytic performance of the Zn modified ZSM-5 is discussed. We paste these here for your check:

“The methanol conversion and product selectivity were calculated as follows:

$$\text{Methanol conversion (\%)} = \frac{n_{\text{CH}_3\text{OH}}^i - n_{\text{CH}_3\text{OH}}^o}{n_{\text{CH}_3\text{OH}}^i} \times 100 \quad (1)$$

The $n_{\text{CH}_3\text{OH}}^i$ and $n_{\text{CH}_3\text{OH}}^o$ represented the total amount of methanol feed and the amount of unreacted methanol, respectively.

$$\text{Product selectivity (\%)} = \frac{m_i}{m} \times 100 \quad (2)$$

The m_i and m represented the mass of product i in the hydrocarbons products and the total mass of all hydrocarbons products, respectively.” (Page 5, Paragraph 3)

“As shown in figure 12B, the lifetime of the catalyst was reduced gradually with the increase of Zn loading. When the active Zn-Lewis acid sites were generated by the impregnation of Zn, the inactive ZnO clusters were also formed which had been reported in our previous studies. [13,44]. These ZnO species dispersed on the surface or in the channels of ZSM-5 zeolites decreased the specific surface area and pore volume. In MTA reaction, the deactivation of the

catalysts was mainly due to the blockage of pore mouth or channel caused by carbon deposition which would inhibit the diffusion of the products and accelerate the deactivation of the catalyst. [65, 66] It was reported that the large ZnO macroparticles, ZnO nanoparticles, and ZnOH⁺ species were found on Zn/HZSM-5 zeolites after the modification of Zn loading. The micron-sized ZnO had little effect on the physicochemical properties of the zeolites, while nano ZnO particles could exist at the pore mouth or inside the channel which was the main reason for the deactivation of Zn modified zeolite catalysts. [61]” (Page 21, Paragraph 2)

Point 6: Discussion on potential approaches to address the fast decay of catalytic performance of Zn modified ZSM-5 should be mentioned at least with cited references.

Response 6: Thanks to the reviewer for pointing out this issue and giving the valuable advices.

With considering the reviewer’s comments, the discussion on potential approaches to address the fast decay of catalytic performance of Zn modified ZSM-5 is mentioned with cited relevant references. We paste these here for your check:

“In order to solve the fast deactivation of catalytic performance of Zn modified ZSM-5 zeolites, the research on the preparation of nano-sized ZSM-5 and mesoporous ZSM-5 zeolites had attracted much attention. It was found that the introducing intracrystalline mesopores into ZSM-5 zeolites significantly improved the stability of the catalysts. [45, 46, 67-69]” (Page 21, Paragraph 2)